# Three-dimensional photonic topological insulator without spin–orbit coupling

Minkyung Kim[1,7], Zihao Wang [2,7], Yihao Yang [2,3,7], Hau Tian Teo [2], Junsuk Rho [1,4,5✉] & Baile Zhang [2,6✉]

Spin–orbit coupling, a fundamental mechanism underlying topological insulators, has been introduced to construct the latter's photonic analogs, or photonic topological insulators (PTIs). However, the intrinsic lack of electronic spin in photonic systems leads to various imperfections in emulating the behaviors of topological insulators. For example, in the recently demonstrated three-dimensional (3D) PTI, the topological surface states emerge, not on the surface of a single crystal as in a 3D topological insulator, but along an internal domain wall between two PTIs. Here, by fully abolishing spin–orbit coupling, we design and demonstrate a 3D PTI whose topological surface states are self-guided on its surface, without extra confinement by another PTI or any other cladding. The topological phase follows the original Fu's model for the topological crystalline insulator without spin–orbit coupling. Unlike conventional linear Dirac cones, a unique quadratic dispersion of topological surface states is directly observed with microwave measurement. Our work opens routes to the topological manipulation of photons at the outer surface of photonic bandgap materials.

[1] Department of Mechanical Engineering, Pohang University of Science and Technology (POSTECH), Pohang 37673, Republic of Korea. [2] Division of Physics and Applied Physics, School of Physical and Mathematical Sciences, Nanyang Technological University, Singapore 637371, Singapore. [3] Interdisciplinary Center for Quantum Information, State Key Laboratory of Modern Optical Instrumentation, ZJU-Hangzhou Global Science and Technology Innovation Center, Zhejiang University, Hangzhou 310027, China. [4] Department of Chemical Engineering, Pohang University of Science and Technology (POSTECH), Pohang 37673, Republic of Korea. [5] POSCO-POSTECH-RIST Convergence Research Center for Flat Optics and Metaphotonics, Pohang 37673, Republic of Korea. [6] Centre for Disruptive Photonic Technologies, Nanyang Technological University, Singapore 639798, Singapore. [7] These authors contributed equally: Minkyung Kim, Zihao Wang, Yihao Yang. ✉email: jsrho@postech.ac.kr; blzhang@ntu.edu.sg

The idea of photonic bandgap was proposed in the 1980s in a 3D artificial material as a means to control photons in much the same way as semiconductor crystals control electrons[1–3]. Due to this similarity, such photonic bandgap materials are often called photonic crystals[4]. By embedding structural defects and light emitters into the bulk of a photonic bandgap material, unprecedented science and technologies of light have been developed, being able to fundamentally engineer light propagation and emission properties with full control[5–10]. Compared to the bulk of photonic bandgap materials, the possibility of manipulating the flow of light at a surface through surface states[11,12], similar to the concept of surface plasmons at a metal surface, is much less explored.

The interest in surface states has risen fast recently due to the discoveries of topological insulators in condensed matter systems[13,14] and the constructions of their photonic analogs, also known as PTIs, in photonic bandgap materials[15–18]. A topological insulator is insulating in the inner bulk due to its energy bandgap, but is conductive on the outer surface through topological surface states that are tied to the nontrivial band topology. When time-reversal symmetry is preserved, all existing topological insulators require a strong spin–orbit coupling to induce band inversion, in order to acquire the nontrivial band topology[13,14]. Moreover, such spin–orbit coupling gives rise to the spin-polarized Dirac surface states at the outer surface of a 3D topological insulator. While many PTIs have been demonstrated for two-dimensional (2D) topological insulators[15–17], so far there is only one experiment demonstrating a 3D PTI with a nontrivial 3D photonic bandgap[19]. However, unlike in the 3D topological insulator, the topological surface states exist only along an internal domain wall between two PTIs with identical but opposite settings.

This discrepancy stems from the intrinsic difference between electrons and photons. Unlike electrons that are fermions, the bosonic nature of photons has excluded the existence of electronic spin, and thus the intrinsic spin–orbit coupling, in photonic systems. As a result, many approaches of constructing the photonic pseudospins have been proposed[15–17]. For example, the in-phase and out-of-phase relation between electric and magnetic fields can be used to define the electron-like spin-up and spin-down states[20]. However, such photonic pseudospins require extra symmetries such as the electromagnetic duality of the unit cell. Since the duality is not preserved at the outer surface, topological Dirac surface states can only emerge along an internal domain wall between two PTIs with identical but opposite settings.

In fact, there is a special topological insulator phase, the topological crystalline insulator[21], which arises from crystal symmetries. Despite the fact that all previous topological crystalline insulators have been realized in materials with spin–orbit coupling[22–26], the original model with C$_4$ symmetry, as proposed by Fu in 2011, does not require spin–orbit coupling, and thus can be treated as the counterpart of conventional topological insulators in materials without spin–orbit coupling[21]. The unique feature for the topological phase of Fu's model is the quadratic dispersion of topological surface states, unlike the conventional Dirac surface states in topological insulators. However, this original Fu's model, despite being discussed widely[21,27–30] has never been realized in any physical platform.

It is worth mentioning that the Fu's model has recently been recognized "fragile[31]" in the sense that its topological states are removable by adding a trivial band in the bandstructure, being less robust than the conventional "stable" topological states that are irremovable by trivial bands. Actually, such fragile topology has been a common practice in 2D photonic topological systems[32–34], such as a few models of 2D PTIs[35,36] that have found promising applications in photonics[37,38]. The recent 3D PTI[19] also belongs to fragile topology[31]. Yet these previous

studies still required the construction of photonic pseudospins. The Fu's model, if realized in photonics, will provide a unique route to get rid of the spin–orbit coupling that is absent in photonics.

Here, we report on the experimental realization of Fu's model in a 3D PTI. The 3D PTI is realized in a 3D photonic bandgap material with C$_4$ symmetry, exhibiting gapless topological quadratic surface states at a symmetry-preserving surface, as predicted in Fu's model. Due to the absence of spin–orbit coupling, there is no need to deliberately construct any photonic pseudospin. Via direct near-field measurements, we experimentally characterize the complete 3D bandgap and map out the self-guided topological surface states, without extra confinement by another PTI or any other cladding. Unlike the well-known Dirac cones of topological surface states, the quadratic dispersion of surface states, as the hallmark feature of the topological crystalline insulator without spin–orbit coupling, is directly observed. Our work demonstrates the unnecessity of spin–orbit coupling in time-reversal-invariant topological insulators and paves the way towards the topological manipulation of photonic surface waves at the outer surface of photonic bandgap materials, which may lead to novel 3D cladding-free topological photonic devices.

## Results and discussion

The unit cell of the designed 3D PTI consists of four metallic split-ring resonators (SRRs) connected to each other (see Fig. 1a). The connected SRRs are invariant under C$_4$ symmetry but have a broken mirror symmetry along the $z$-axis. The background material (F4B-255) has a dielectric constant $2.55 \pm 0.05$. The unit cell is arranged periodically in a 2D square lattice in the $x$–$y$ plane and is stacked along the $z$-axis. Following the same symmetries as in Fu's model[21], this structure has a space group of $\mathcal{T}_3 \times C_{4v} \times \mathbb{Z}_2^T$, where $\mathcal{T}_3$ represents a 3D translational group, C$_{4v}$ is a point group generated by C$_4$ and a vertical mirror plane, and $\mathbb{Z}_2^T$ denotes the order-two group generated by time-reversal symmetry. The PTI has a complete bandgap above the first and second bands (see Fig. 1c). The topological surface states of the PTI are expected to appear at a symmetry-preserving surface, i.e., (001) surface (Fig. 1b), which has a reduced spatial symmetry of $\mathcal{T}_2 \times C_{4v} \times \mathbb{Z}_2^T$. Particularly, as predicted in Fu's model, at the C$_4$-invariant point $\bar{M}$ is a quadratic band crossing of the gapless surface states, unlike the conventional Dirac cones of surface states as in 3D topological insulators.

Geometrical parameters of the connected SRRs and their periodicity are optimized by using a particle swarm optimization to broaden the bandgap size (see "Methods"). The mode analysis shows that electric field profiles of the three lowest bands are reminiscent of the electronic orbitals, $p$ orbitals for the first and second bands, and $d$ orbital for the third band (Fig. 1d). The three bands are labeled as $p_{x=-y}$, $p_{x=y}$, and $d_{xy}$ modes in order. Importantly, the first and second bands that are degenerate at $\bar{M}$ as a result of the C$_4$ symmetry have mutually orthogonal linear polarizations along $x = -y$ and $x = y$ (see Fig. 1d, i and ii). The two orthogonal $p$ orbital-like modes form exact analogies of $p_x$ and $p_y$ orbitals in Fu's model[21] (see Supplementary Note 1). The orbital degree of freedom that replaces the spin degree of freedom is a key to the topological crystalline insulator phase without spin–orbit coupling.

Next, we fabricate the experimental sample of the 3D PTI by stacking 30 layers of printed circuit boards (PCBs) containing $31 \times 31$ unit cells per layer. Each layer is paired with a bare board with thickness of 6 mm as a spacer (Fig. 2a). All boards have air holes at the center of unit cells to allow a probe tip to be inserted inside the bulk sample. To measure the bulk transmission as well as bandstructure, a dipole source antenna is placed near the

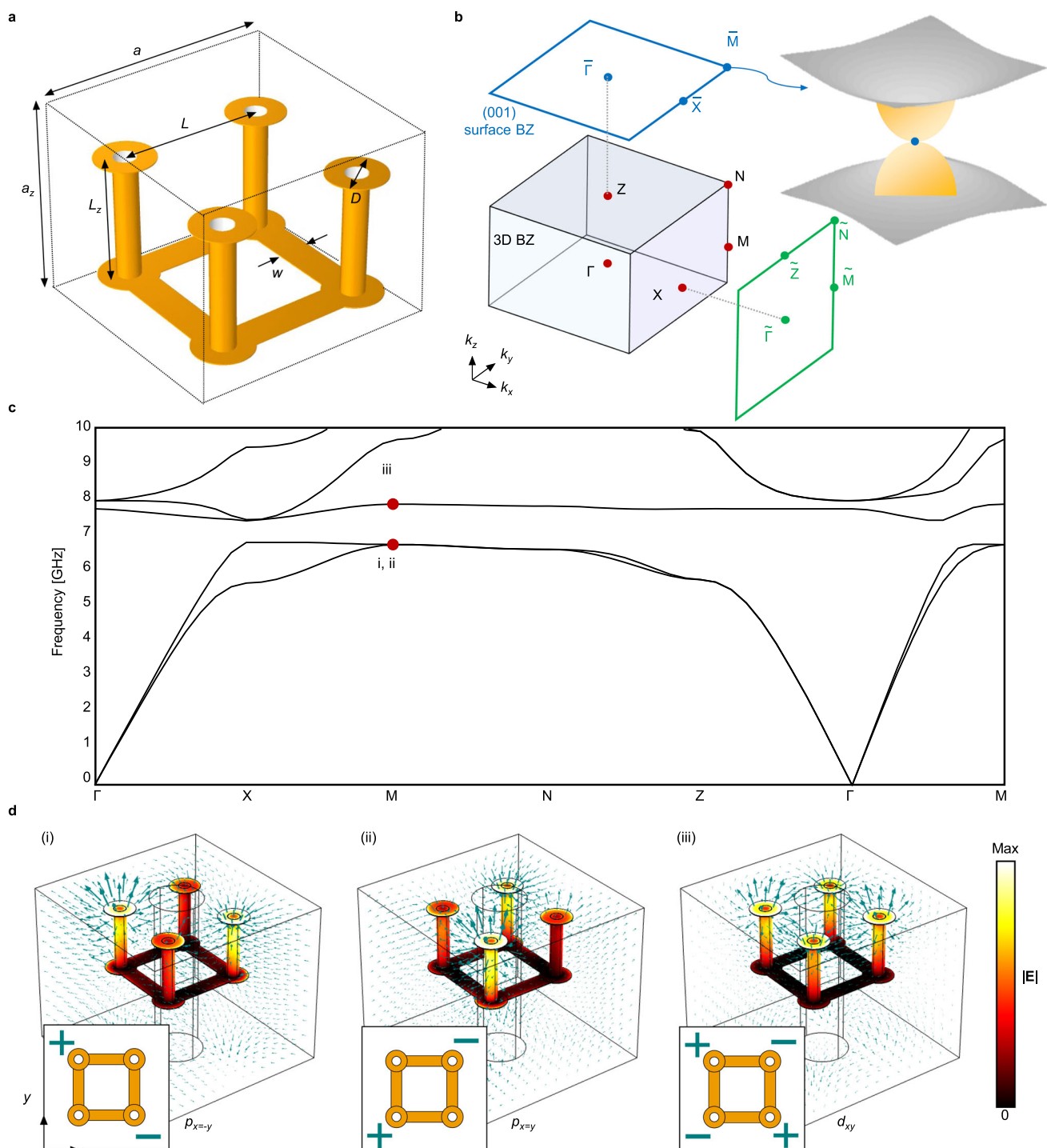

**Fig. 1 The 3D PTI without spin–orbit coupling. a** Unit cell of the 3D PTI. The geometrical parameters are given as $a = 11$ mm, $a_z = 9.7$ mm, $L = 4.4$ mm, $L_z = 3.7$ mm, $w = 1$ mm, and $D = 1.8$ mm. The golden region is copper. **b** The 3D BZ and its projection to the $k_x$- and $k_z$-axes. Inset: a quadratic band crossing at a $C_4$-invariant momentum $\bar{M}$ in the (001) surface BZ. **c** Simulated bulk bandstructure. **d** Electric field distribution (colormap) and polarization density (arrows) of bulk eigenmodes at $\bar{M}$ for the (i) first, (ii) second, and (iii) third bands. + and − indicate where electric fields are divergent and convergent, respectively.

bottom center of the sample, then a dipole probe antenna captures the spatial distribution of excited bulk modes in the $y$–$z$ plane (dashed box) inside the air holes (see "Methods"). We first probe the transmittance spectrum at the location approximately 10 unit cells away from the source position (see Fig. 2b, yellow marker). As the dipole source antenna can excite the bulk states with arbitrary wavevectors, the measured spectrum proves a

complete 3D bandgap between 6.82 and 7.42 GHz, with a relative bandgap of 8.4% (see Fig. 2c). We then scan the complex field distribution on the middle $y$–$z$ plane point by point (see Fig. 2b). Applying the Fourier transform to the measured field profiles, we obtain the projected bulk bandstructure (see Fig. 2d). One can see a complete bandgap for all wavevectors, consistent with the bulk transmission measurement. Besides, we also compare the

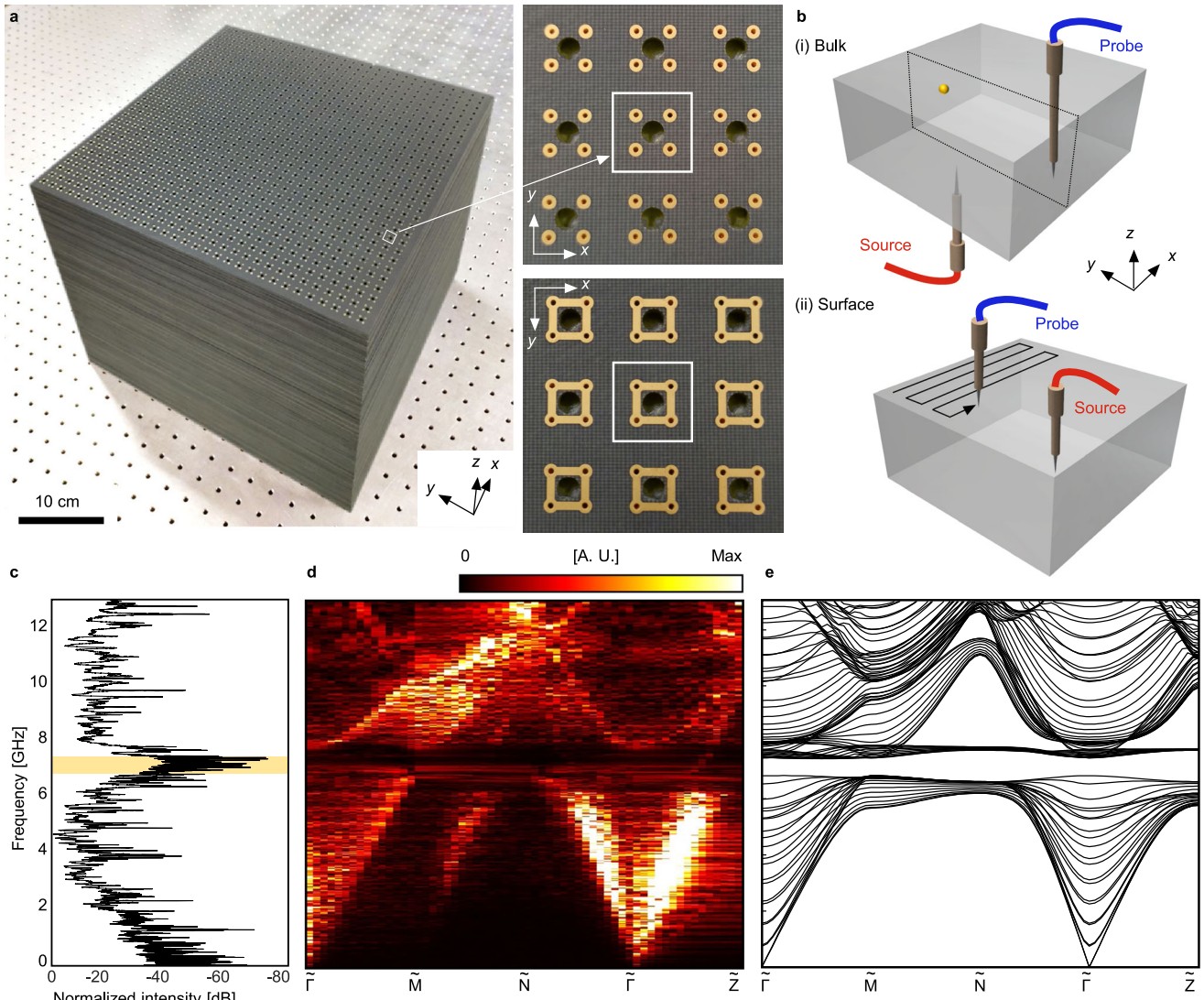

**Fig. 2 Experimental setup and measurement of the bulk dispersion. a** Photograph of the sample. Right insets denote magnified images of top and bottom sides of the PCBs. **b** Experimental setups for (i) bulk and (ii) surface dispersion measurements. For bulk dispersion measurement, a dipole source antenna is placed at the bottom near the middle $y$–$z$ plane (dashed box), and a dipole probe antenna is inserted into the sample to scan the middle $y$–$z$ plane. For surface dispersion measurement, the probe antenna scans the top surface, while a dipole source antenna is placed slightly above the corner of the surface. **c** Normalized transmittance measured at the yellow marker in (**b**). Yellow region indicates a bulk bandgap. Measured (**d**) and simulated (**e**) bulk dispersions of the 3D PTI.

measured dispersion with the simulated counterpart and find an excellent agreement between them (Fig. 2e).

Then, we perform experiments to measure the in-gap topological surface states on the (001) surface of the 3D PTI. We directly scan the top surface of the 3D PTI without using any cladding. The source is placed at a corner of the top surface (see Fig. 2b, ii). By applying a similar procedure as the bulk measurement, we obtain the surface dispersion. Figure 3a shows the surface dispersion along the high symmetry lines. One can see that in-gap surface states connect the lower and upper bulk bands, and a quadratic band touching point occurs at the high-symmetric point $\bar{M}$. The measured isofrequency contours at 6.59, 6.76, and 7.09 GHz (see Fig. 3d) further confirm the quadratic dispersion centered at $\bar{M}$. The experimental results are consistent with the numerical ones (see Fig. 3b, c). Such single quadratic surface states at $\bar{M}$ are the hallmark experimental evidence of the Fu's model of topological crystalline insulator without spin–orbit coupling[21]. Note that the black lines in Fig. 3b denote the light cone. The topological surface states carry high momenta and are

self-guided below the light cone. The surface states near the light cone are spoof surface plasmon polaritons (SPPs) that originate from the periodic metallic structure itself[39]. These states are weakly confined at both top and bottom surfaces and have no topological origin. The quadratic surface states hybridize with the spoof SPPs when their spatial momenta are close to each other. As a result, the dispersion branch along the $\bar{M}$-$\bar{\Gamma}$ line does not connect to the upper band, but connects to the dispersion of spoof SPPs (Fig. 3b). This hybridization does not affect the quadratic degenerate point at $\bar{M}$.

The existence of the surface states and their quadratic dispersion are protected topologically by the $C_4$ and time-reversal symmetries[21]. The $C_4$ symmetry enforces the overlap of two vertices of the projected Brillouin zone (BZ) at $\bar{M}$ and results in one quadratic dispersion instead of two linear dispersions. Due to the $C_4$ symmetry required, the topological surface states only appear at the symmetry-preserving (001) surface. Band topology of the 3D PTI can be further confirmed by examining the spectral flow of Zak phases[28] of the two bulk bands below the bandgap

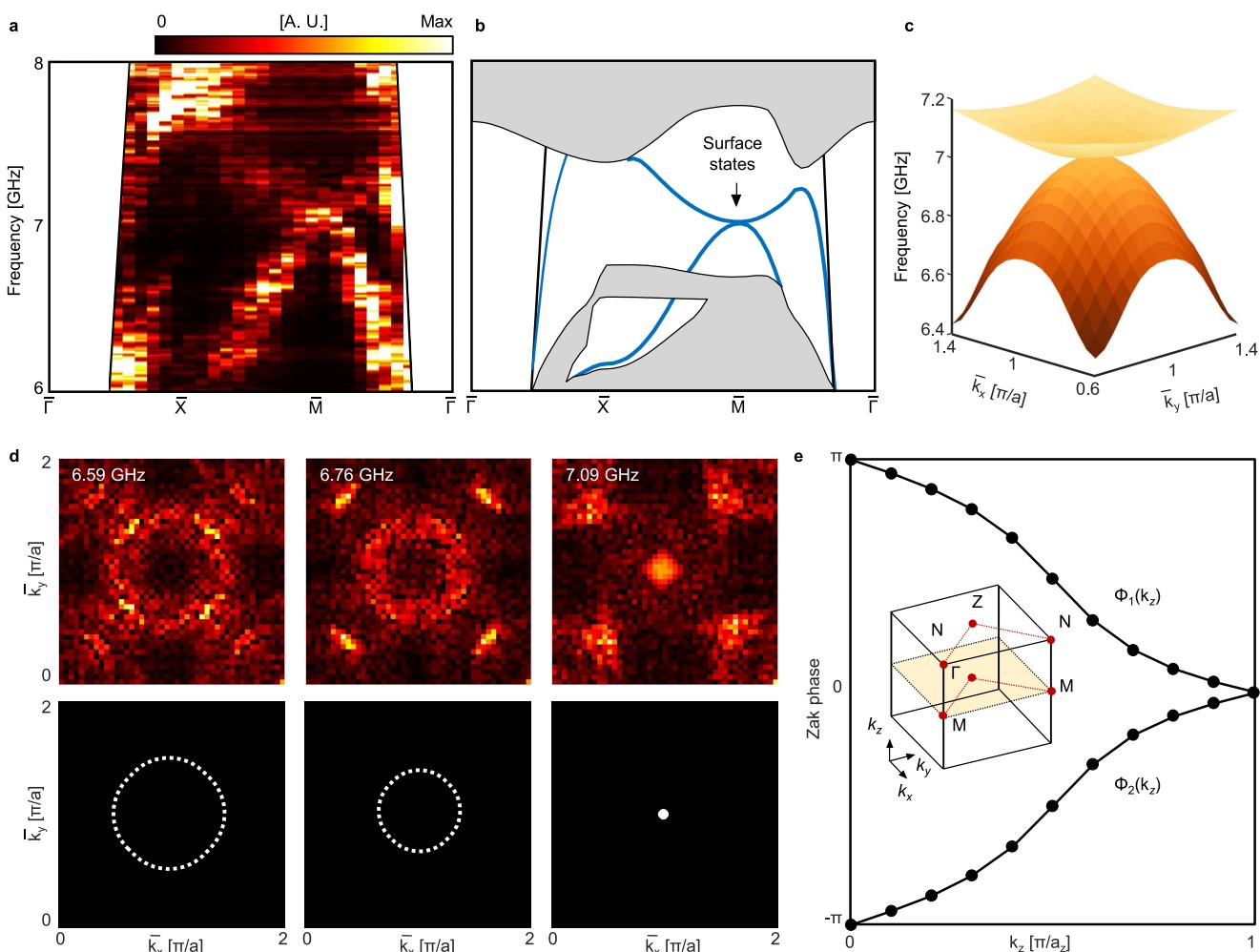

**Fig. 3 Topological quadratic surface states on the (001) surface of the 3D PTI. a, b** Measured (**a**) and simulated (**b**) surface dispersions. Gray areas represent the projected bulk bands and blue curves denote surface states localized at the top surface. Black lines indicate the light cone. **c** Simulated surface dispersion in the (001) surface BZ. **d** Measured (top) and simulated (bottom) isofrequency contours of the surface states at 6.59, 6.76, and 7.09 GHz. **e** Spectral flow of the Zak phases. Inset shows the 3D BZ with a family of $C_4$-invariant loops, along which the Zak phases are calculated. The Zak phase diagram proves the nontrivial $\mathbb{Z}_2$ topological invariant, $v_0 = 1$.

using a Wilson loop method[40]. The Zak phases at a given $k_z$ denote the geometric phase acquired while taking a $C_4$-invariant loop in the constant $k_z$ plane for $k_z \in [0, \pi/a_z]$ (Fig. 3e, inset). Two Zak phases, $\Phi_1(k_z)$ and $\Phi_2(k_z)$, are computed for this rank-2 band (see "Methods"). Because of the symmetry conditions ($C_4$ and time-reversal symmetries), the Zak phases obey $\Phi_1(k_z) = -\Phi_2(k_z)$ (mod $2\pi$) for all $k_z$ and are quantized to 0 or $\pi$ (mod $2\pi$) at $k_z = 0$ and $k_z = \pi/a_z$ plane[31]. Importantly, the Zak phase at the two planes are distinct to each other, that is, $\Phi_1(k_z = 0) \neq \Phi_1(k_z = \pi/a_z)$. The spectral flow of the Zak phase, i.e., interpolation of the Zak phase across its maximally allowed ranges $(-\pi, \pi]$, is a definite sign of 3D topological crystalline insulator phase[21]. The topological invariant of the corresponding Zak phases is nontrivial, $v_0 = 1$, which belongs to the strong topological phase and is consistent with the existence of the quadratic surface states[21,28].

We have thus provided the experimental characterization of a 3D PTI in a $C_4$-symmetric and time-reversal-invariant photonic bandgap material. We experimentally observe both the 3D complete bandgap and the single quadratic band crossing of the gapless surface states at the BZ corner, as the hallmark signature of Fu's original model of topological crystalline insulator[21]. Nontrivial spectral flow of the Zak phases from the first-

principles calculation also confirms the topological phase. The demonstrated unnecessity of spin–orbit coupling in the design and construction of 3D PTIs opens new venues in the topological manipulation of photonic surface states at the outer surface of photonic bandgap materials. Although our work was performed in the microwave regime, the design and approach can be generalized to higher frequencies such as terahertz or infrared regime[41], which will enable topologically robust photonic devices in cladding-free 3D geometries.

## Methods

**Details of numerical simulation and optimization.** All numerical band dispersions are obtained by using a finite element method-based software (COMSOL Multiphysics 5.5, eigenfrequency solver). Bulk dispersion is simulated by using a unit cell with Bloch boundary conditions along all boundaries. For surface dispersion, a supercell that consists of 9 unit cells is used with Bloch boundary conditions along the x- and y-axes and perfectly matched layer conditions along the z-axis with air spacing.

The unit cell design is optimized to maximize the relative bandgap, which is defined as a ratio of bandgap width to the bandgap center frequency. More specifically, the relative bandgap is obtained by $g = (f_{up} - f_{down})/(f_{up} + f_{down}) \times 2$ for $f_{up}$ being the minimum frequency of the third band and $f_{down}$ being the maximum frequency of the second band. In a particle swarm optimization, five parameters, $a_z$, $L$, $L_z$, $w$, and $D$, are optimized to yield minimum $f = 1/g^2$ by linking COMSOL Multiphysics 5.5 and MATLAB via Livelink for MATLAB. Bulk

dispersion of a unit cell along $\Gamma$-$M$-$N$-$Z$-$\Gamma$-$M$ is simulated iteratively using updated geometrical parameters. The iteration is repeated a hundred times with ten populations per iteration.

**Near-field scanning measurement**. The near-field spectra are experimentally measured with a vector network analyzer (R&S ZNB20). Two measurements are performed to obtain the projected bulk bands and surface bands, respectively. To measure the projected bulk bands, the source is placed at the center position at the bottom of the whole sample. The probe is inserted into the sample through the air holes to collect the electric fields in each unit cell in the $y$–$z$ plane, which is in the middle of the $x$-direction. For the surface state measurement, a 3 mm-thick bare board with air holes is placed on the top of the whole sample. The source is fixed on the corner of the top surface (1 mm) and the probe scans the top $x$–$y$ plane. Because the resolution of the measured surface dispersion is determined by the number of unit cells of the sample, we apply $C_4$ operations to the spatial field profile to artificially enlarge the measurement area and to double the resolution (Supplementary Fig. 1). This allows us to employ three $C_4$ operations to quadruple the measured area, in which the source is placed at a center. In the surface state measurement, some absorbers are also used around the source to prevent the waves from being coupled to the air. In both measurements, four lateral sides of the sample are covered by the microwave absorbers.

**Zak phase calculation**. The Wilson loop is given as[31] $\mathscr{W}_{pq}(C) = \mathscr{P}\exp[i\oint_C \langle u_{qk}|i\nabla_k|u_{pk}\rangle \cdot \mathrm{d}k]$, where $C$ is the $C_4$-invariant loop ($M$-$\Gamma$-$M$ for $k_z = 0$ and $N$-$Z$-$N$ for $k_z = \pi/a_z$), $\mathscr{P}$ is a path-ordered integration, $p$ and $q$ are indices of bands of interest, $u_{pk}$ is the Bloch function of the $p$-th band at $k$, and the bracket integrates the inner expression over a unit cell. Note that for the rank-$n$ band, the Wilson loop is a $n$ by $n$ matrix. To eliminate the gauge dependency and to enable computation from discretized data, $\mathscr{W}_{pq}(C)$ is calculated in terms of the path-ordered products of band projections[28,31]. The 3D PTI reported here has two bands below the bandgap ($p$, $q \in \{1, 2\}$) and thus has two eigenvalues of the Wilson loop $\{\exp(i\Phi_1(k_z)), \exp(i\Phi_2(k_z))\}$. Arguments of the eigenvalues are Zak phases, $\Phi_1(k_z)$ and $\Phi_2(k_z)$. For more than ten momentum points in the $C_4$-invariant loops, the Zak phases converge well (Supplementary Fig. 7).

$\mathbb{Z}_2$ topological invariant $\nu_0$ is defined as[21]

$$(-1)^{\nu_0} = (-1)^{\nu_{\mathrm{TM}}}(-1)^{\nu_{AZ}}, \tag{1}$$

where terms on the right-hand side are defined as

$$(-1)^{\nu_{k_1 k_2}} = \exp\left(i\int_{k_1}^{k_2} A_k \cdot \mathrm{d}k\right)\frac{\mathrm{Pf}[w(k_2)]}{\mathrm{Pf}[w(k_1)]}, \tag{2}$$

and $A_k = -\sum_j \langle u_j(k)|i\nabla_k|u_j(k)\rangle$ is the Berry connection, Pf denotes Pfaffian, and

$$w_{mn}(k) = \langle u_m(k)|UT|u_n(k)\rangle, \tag{3}$$

for time-reversal operator $T$ and $C_4$ rotation operator $U$.

## Data availability
The data that support the findings of this study are openly available in NTU research data repository DR-NTU (Data) at https://doi.org/10.21979/N9/C05UAS.

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

## Acknowledgements
This work was financially supported by the POSCO-POSTECH-RIST Convergence Research Center program funded by POSCO, and the National Research Foundation

(NRF) grant (NRF-2019R1A2C3003129) funded by the Ministry of Science and ICT (MSIT) of the Korean government. M.K. acknowledges the NRF *Sejong* Science fellowship (NRF-2022R1C1C2004662) funded by the MSIT of the Korean government. B.Z. acknowledges support by Singapore Ministry of Education Academic Research Fund Tier 3 Grant No. MOE-2016-T3-1-006, Tier 2 Grant No. MOE-2018-T2-1-022(S), and by Singapore National Research Foundation Competitive Research Program Grant no. NRF-CRP23-2019-0007. The work at Zhejiang University was sponsored by the National Natural Science Foundation of China (NSFC) under Grants No. 62175215.

## Author contributions

J.R., M.K., B.Z., and Y.Y. conceived the idea and initiated the project. M.K. and Y.Y. designed the photonic topological insulator. M.K. performed numerical simulations and built analytical modeling. Z.W. and H.T.T. conducted measurements and analyzed data. M.K., Y.Y., B.Z., and J.R. wrote the manuscript. B.Z. and J.R. supervised the work.

## Competing interests

The authors declare no competing interests.
