## [Peer review file · Nature Communications]

REVIEWERS' COMMENTS

Reviewer #1 (Remarks to the Author):

The revised paper is ready for publication. To the best of my knowledge, this is the first paper to demonstrate PTIs whose topological surface states emerge really at a surface, without any cladding, and it realizes experimentally the so-called Fu model that was proposed a decade ago but it was never realized in experiment.

Reviewer #3 (Remarks to the Author):

I am grateful to the authors for adding a discussion about the fragile topology in the introduction and citing the references I suggested. I found the author's response comprehensive and convincing. The realization of photonic topological insulators with the fragile topology would be meaningful scientifically and technologically. I also found the addition of supplementary notes welcome. Furthermore, as noted in my original report, the result is appropriate for publication in Nature Communications. For these reasons, I believe that the article deserves publication in the journal.

Response Letter to Reviewers

COMMENTS FROM 1st REVIEWER:

The revised paper is ready for publication. To the best of my knowledge, this is the first paper to demonstrate PTIs whose topological surface states emerge really at a surface, without any cladding, and it realizes experimentally the so-called Fu model that was proposed a decade ago but it was never realized in experiment.

Response from Authors:

We thank the reviewer for the strong support and for acknowledging the novelty of our work.

COMMENTS FROM 3rd REVIEWER:

I am grateful to the authors for adding a discussion about the fragile topology in the introduction and citing the references I suggested. I found the author's response comprehensive and convincing. The realization of photonic topological insulators with the fragile topology would be meaningful scientifically and technologically. I also found the addition of supplementary notes welcome. Furthermore, as noted in my original report, the result is appropriate for publication in Nature Communications. For these reasons, I believe that the article deserves publication in the journal.

Response from Authors:

We are grateful to the reviewer for acknowledging our efforts during the revision and providing strong support.